# Chemical Constituents from *Fraxinus hupehensis* and Their Antifungal and Herbicidal Activities

**DOI:** 10.3390/biom10010074

**Published:** 2020-01-02

**Authors:** Chi-Na Zhao, Zong-Li Yao, Dan Yang, Jian Ke, Qing-Lai Wu, Jun-Kai Li, Xu-Dong Zhou

**Affiliations:** 1School of Agriculture, Yangtze University, Jingzhou 434020, China; zhaochina0623@163.com (C.-N.Z.); y806032526@163.com (Z.-L.Y.); jin53896@163.com (D.Y.); jk2427109364@163.com (J.K.); 2Institute of Pesticides, Yangtze University, Jingzhou 434020, China; 3TCM and Ethnomedicine Innovation & Development Laboratory, School of Pharmacy, Hunan University of Chinese Medicine, Changsha 410208, Hunan, China

**Keywords:** *Fraxinus hupehensis*, isolation and characterization, phytochemical investigation, fungicide, herbicidal activity

## Abstract

The phytochemical investigation of *Fraxinus hupehensis* led to the isolation and characterization of ten compounds which were identified as fraxin (**1**), fraxetin (**2**), esculetin (**3**), cichoriin (**4**), euphorbetin (**5**), kaempferol-3-*O*-*β*-rutinoside (**6**), oleuropein (**7**), linoleic acid (**8**), methyl linoleate (**9**), and *β*-sitosterol (**10**). Structures of the isolated constituents were characterized by ^1^H NMR, ^13^C NMR and HRMS. All the compounds, except compounds **3** and **4,** were isolated for the first time from this plant. Further, this was the first report for the occurrence of compound **5** in the *Fraxinus* species. Antifungal activity evaluation showed that compound **2** exhibited significant inhibitory effects against *Bipolaris maydis, Sclerotium rolfsii,* and *Alternaria solani* with EC_50_ values of 0.31 ± 0.01 mmol/L, 10.50 ± 0.02 mmol/L, and 0.40 ± 0.02 mmol/L respectively, compared to the positive control, Carbendazim, with its EC_50_ values of 0.74 ± 0.01 mmol/L, 1.78 ± 0.01 mmol/L and 1.41 ± 0.00 mmol/L. Herbicidal activity tests showed that compounds **8**–**10** had strong inhibitory effects against the roots of *Echinochloa crus-galli* with EC_50_ values of 1.16 ± 0.23 mmol/L, 1.28 ± 0.58 mmol/L and 1.33 ± 0.35 mmol/L respectively, more potently active than that of the positive control, Cyanazine, with its EC_50_ values of 1.56 ± 0.44 mmol/L. However, none of the compounds proved to be active against the tested bacteria (*Erwinia carotovora, Pseudomonas syringae,* and *Ralstonia solanacearum*).

## 1. Introduction

The genus *Fraxinus* includes more than 60 species in the world and 30 native species in China, many of which commonly used in Traditional Chinese Medicines. *F. hupehensis*, a member of Oleaceae family, is only distributed in Zhongxiang and Jingshan counties, Hubei province of China [1]. It grows slowly, lives long and often forms spines in pairs on their branches. Besides, due to its beautiful tree shape and twisted roots, it is considered to be an ideal bonsai and landscape greening plant which has been called *Living fossil* or *King of bonsai* [2]. In folk applications, it is very popular because of its delicate appearance and its lack of pests [3]. However, since it was discovered as a new species in 1979, *F. hupehensis* has been subjected to excessive resource exploitation. In addition, its seeds have long dormancy period (>1 year) which limits their dispersal, so the populations of this precious plant have been continuing to decline, and it was officially recognized as a rare and endangered plant in China in 1990 [4].

In traditional Chinese medicine, some species of the genus *Fraxinus* have been used and the barks of them called *Cortex Fraxini* [5], commonly known as ‘Qin-Pi’ in Chinese. At present, the origin of Chinese medicine ‘Qin-Pi’ mainly includes *F. rhynchophylla* Hance, *F. chinensis* Roxb, *F. szaboana* Lingelsh and *F. stylosa* Lingelsh. The bark of *F. hupehensis* is sometimes used as *Cortex Fraxinus* [6]. Previous studies have shown that they have anti-inflammatory [7], anti-tumor [8], anti-bacterial [9], anti-viral [10], and antioxidant [11] activities. In addition, some literature indicated the leaves extract exhibited antifungal activities [12,13,14]. However, there are only two previously reports [15,16] for the phytochemical investigation of *F. hupehensis* to the best of our knowledge. Based on the above information, it is necessary to systematically study its phytochemistry and examine its fungicidal, herbicidal and bactericidal activities.

## 2. Experimental

### 2.1. Plant Collection and Authentication

The leaves and bark of *F. hupehensis* were collected in the Bonsai garden of the western campus of Yangtze University, Hubei, China in May 2017, and identified by one of authors, Prof. Qing-Lai Wu. A voucher specimen was deposited in the herbarium of Yangtze University (YZU201705-FH).

### 2.2. Extraction, Isolation and Identification

The leaves of *F. hupehensis* (10.0 kg) were dried at 25 °C and pulverized by a plant grinder (Shanghai Heysu Pharmceutical Machinery co., LTD, Shanghai, China). They were extracted three times with 60 L of 80% MeOH at room temperature. After removal of the solvent under reduced pressure, the residue of crude extract was gained (1.02 kg). The crude extract was suspended in H_2_O (1.5 L) and partitioned successively with petroleum ether, EtOAc and n-butanol, respectively. The EtOAc extract (180 g, solid) was subjected to a silica gel column chromatography (CC), and eluted with CHCl_3_-MeOH (20:1, 18:1, 16:1, 14:1, 12:1, 10:1, 8:1, 6:1, 4:1, 2:1, 1:1) to obtain 26 fractions (A–Z). Fraction F was applied to silica gel CC and eluted with a gradient of chloroform-methanol (90:10–1:100) to give 18 sub-fractions (F1–F18). The fraction F1 was dissolved in a solvent of chloroform-methanol (4:1) and recrystallized at −10 °C to obtain compound **1**. Fraction G was dissolved in methanol and filtered. The filtrate was evaporated, and then chromatographed on a silica gel CC with CHCl_3_-MeOH (20:1~5:1) to yield five sub-fractions (G1–G5). The sub-fraction G5 was recrystallized to give compounds **2** and **3**. Fraction O was chromatographed on a silica gel CC eluting with CHCl_3_-MeOH (5:1~2:1) to obtain three sub-fractions (O1–O3). And then the O2 part was recrystallized to give compound **4**. Fraction L was subjected to a Sephadex LH–20 CC (Pharmacia Biotech Ltd. Piscataway, NJ, USA), eluting with MeOH to yield compound **7**. Fraction P was subjected to C_18_ reverse phase CC and eluted with MeOH-H_2_O (50:50–100:0) to gain ten sub-fractions (P1–P10). Sub-fraction P6 was separated by Sephadex LH–20 CC (MeOH) to give compound **6**. Sub-fraction P8 was purified by C_18_ reverse column CC, eluting with MeOH-H_2_O (70:30) to give compound **5**.

The bark of *F. hupehensis* (1 kg) was extracted with 95% EtOH twice, each time at 80 °C for 3 h. The EtOH extract was isolated on a silica gel column to give seven fractions (Fr.1–7), and Fr.1 was repeatedly purified by Sephadex LH–20 CC, eluting with CHCl_3_-MeOH (1:2) to give compound **8**. Fr.2 was separated by silica gel CC (200–300 mesh) to obtain eight sub-fractions (Fr.2.1–2.8). Fr.2.3 was subjected on a Sephadex LH–20 CC to give compound **9**. Fr.2.5 was recrystallized three times to give compound **10** (See Figure 1).

Chemicals and solvents were purchased from commercial suppliers and were used without further purification. Thin-layer chromatography (TLC) was performed on silica gel 60 F254 (Qingdao Marine Chemical Ltd., Qingdao, China). Column chromatography (CC) was performed over silica gel (200–300 mesh, Qingdao Marine Chemical Ltd.). ^1^H and ^13^C NMR spectrum were recorded in CDCl_3_ or DMSO-*d*_6_ solution on a Bruker 400 MHz spectrometer (Bruker Co., Fällanden, Switzerland), using tetramethyl silane (TMS) as an internal standard, and chemical shift values (*δ*) were given in parts per million (ppm). The following abbreviations were used to designate chemical shift multiplicities: s = singlet, d = doublet, t = triplet, q = quartet, m = multiple. MS data were obtained using an APEX IV Fourier-Transform Mass Spectrometry (Bruker). All spectra are shown in Appendix A.

Compound **1**: Yellow solid; m.p. 145–146 °C; HRESIMS: *m*/*z* 369.0827 [M − H]^−^ (calcd for C_16_H_17_O_10_, 369.0822). ^1^H NMR (DMSO-*d_6_*; 400 MHz): *δ* ppm 7.93 (d, *J* = 9.6 Hz, 1H), 7.07 (s, 1H), 6.27 (d, *J* = 9.6 Hz, 1H), 5.18 (d, *J* = 4.7 Hz, 1H), 5.03 (d, *J* = 4.7 Hz, 1H), 4.97 (d, *J* = 7.7 Hz, 1H), 3.14–3.44 (overlapped ), 3.81 (s, 6–OCH_3_, 3H). ^13^C NMR (DMSO-*d_6_*; 101 MHz): *δ* ppm 160.66, 145.88, 145.22, 144.13, 143.13, 132.01, 112.70, 110.59, 105.39, 104.34, 77.81, 76.70, 74.31, 70.07, 61.21, 56.54. Comparing these NMR data with ref. [17], compound **1** was identified as fraxin.

Compound **2**: Yellow solid; m.p. 228–229 °C; HRESIMS: *m*/*z* 209.0438 [M + H]^+^ (calcd for C_10_H_9_O_5_, 209.0450). ^1^H NMR (DMSO-*d_6_*; 400 MHz): *δ* ppm 9.52 (s, 2H), 7.89 (d, *J* = 9.6 Hz, 1H), 6.80 (s, 1H), 6.22 (d, *J* = 9.6 Hz, 1H), 3.86 (s, 3H). ^13^C NMR (DMSO-*d_6_*; 101 MHz): *δ* ppm 160.99, 145.80, 145.50, 139.77, 139.76, 133.31, 112.30, 110.70, 100.77, 56.50. Comparing these data with ref. [17], compound **2** was identified as fraxetin.

Compound **3**: Yellow solid; m.p. 271–273 °C; HRESIMS: *m*/*z* 179.0333 [M + H]^+^ (calcd for C_9_H_7_O_4_, 179.0344). ^1^H NMR (DMSO-*d_6_*; 400 MHz): *δ* ppm 9.83 (s, 2H), 7.87 (d, *J* = 9.6 Hz, 1H), 6.98 (s, 1H), 6.75 (s, 1H), 6.17 (d, *J* = 9.6 Hz, 1H). ^13^C NMR (DMSO-*d_6_*; 101 MHz): *δ* ppm 161.25, 150.85, 148.95, 144.89, 143.33, 112.78, 111.96, 111.22, 103.10. Comparing these data with ref. [17], compound **3** was identified as esculetin.

Compound **4**: White solid; m.p. 196–198 °C; HRESIMS: *m*/*z* 341.0862 [M + H]^+^ (calcd for C_15_H_17_O_9_, 341.0873). ^1^H NMR (DMSO-*d_6_*; 400 MHz): *δ* ppm 9.00 (s, 1H), 7.93 (d, *J* = 9.5 Hz, 1H), 7.14 (s, 1H), 7.10 (s, 1H), 6.31 (d, *J* = 9.5 Hz, 1H), 5.38 (s, 1H), 5.12 (d, *J* = 3.3 Hz, 1H), 5.10 (d, *J* = 5.1 Hz, 1H), 4.94 (d, *J* = 7.3 Hz, 1H), 4.66 (t, *J* = 5.3 Hz, 1H), 3.75 (dd, *J* = 9.4, 4.8 Hz, 1H), 3.46 (q, *J* = 5.7 Hz, 3H), 3.39–3.30 (m, 1H), 3.17 (d, *J* = 4.1 Hz, 1H). ^13^C NMR (DMSO-*d_6_*; 101 MHz): *δ* ppm 161.01, 149.27, 148.26, 144.62, 144.04, 113.91, 113.43, 113.10, 103.84, 101.46, 77.74, 76.33, 73.64, 70.27, 61.20. Comparing these data with ref. [17], compound **4** was identified as cichoriin.

Compound **5**: Orange oil; HRESIMS: *m*/*z* 353.03.3 [M − H]^−^ (calcd for C_18_H_9_O_8_, 353.0297). ^1^H NMR (DMSO-*d_6_*; 400 MHz): *δ* ppm 9.75 (s, 4H), 7.14 (d, *J* = 9.4 Hz, 2H), 6.89 (s, 2H), 6.14 (dd, *J* = 9.1 Hz, 2H). ^13^C NMR (DMSO-*d_6_*; 101 MHz): *δ* ppm 161.02, 150.75, 149.24, 143.22, 141.55, 118.13, 112.04, 110.81, 102.55. Comparing these data with ref. [18], compound **5** was identified as euphorbetin.

Compound **6**: Yellow solid; m.p. 181–183 °C; HRESIMS: *m*/*z* 593.1515 [M − H]^–^ (calcd for C_27_H_29_O_15_, 693.1506). ^1^H NMR (DMSO-*d_6_*; 400 MHz): *δ* ppm 12.56 (s, 1H), 7.99 (d, *J* = 8.8 Hz, 2H), 6.88 (d, *J* = 8.8 Hz, 2H), 6.41 (d, *J* = 2.0 Hz, 1H), 6.20 (dd, *J* = 2.0 Hz, 1H), 5.31 (d, *J* = 7.4 Hz, 1H), 4.38 (s, 1H), 4.03 (q, *J* = 7.1 Hz, 1H), 3.69 (d, *J* = 9.7 Hz, 1H), 3.63–2.89 (overlapped, 9H), 0.97 (d, *J* = 6.1 Hz, 3H),. ^13^C NMR (DMSO-*d_6_*; 101 MHz): *δ* ppm 177.80, 164.70, 161.65, 160.36, 157.26, 156.98, 133.66, 131.34, 131.34, 121.35, 115.56, 104.34, 101.82, 101.24, 99.26, 94.25, 76.81, 76.19, 74.64, 72.27, 71.05, 70.81, 70.37, 68.71, 60.23, 18.2. Comparing these data with ref. [19], compound **6** was identified as kaempferol–3–rutinoside.

Compound **7**: Yellow solid; m.p. 89–90 °C; HRESIMS: *m*/*z* 539.1775 [M − H]^–^ (calcd for C_25_H_31_O_13_, 539.1765). ^1^H NMR (DMSO-*d_6_*; 400 MHz): *δ* ppm 8.74 (s, 2H), 7.52 (d, *J* = 2.0 Hz, 1H), 6.64 (d, *J* = 8.0 Hz, 1H), 6.61 (d, *J* = 2.0 Hz, 1H), 6.48 (dd, *J* = 8.0, 2.0 Hz, 1H), 5.97 (q, *J* = 6.9 Hz, 1H), 5.87 (t, *J* = 1.8 Hz, 1H), 5.28–4.89 (m, 3H), 4.65 (d, *J* = 7.8 Hz, 1H), 4.51 (s, 1H), 4.08 (qt, *J* = 10.7, 7.2 Hz, 2H), 3.86 (dd, *J* = 9.1, 4.2 Hz, 1H), 3.70 (m, 1H), 3.65 (s, 3H), 3.46 (dd, *J* = 11.7, 6.3 Hz, 1H), 3.25–3.00 (m, 2H), 3.08 (td, *J* = 8.9, 4.2 Hz, 2H), 2.68 (t, *J* = 7.2 Hz, 2H), 2.50 (t, *J* = 1.9 Hz, 1H), 2.40 (dd, *J* = 14.4, 9.3 Hz, 1H), 1.65 (dd, *J* = 7.1, 1.4 Hz, 3H). ^13^C NMR (DMSO-*d_6_*; 101 MHz): *δ* ppm 171.15, 166.65, 153.90, 145.55, 144.22, 129.60, 128.86, 123.51, 120.01, 116.64, 115.99, 108.15, 99.45, 93.39, 77.82, 76.97, 73.73, 70.39, 65.52, 61.56, 51.73, 49.07, 34.16, 30.59, 13.47. Comparing these data with ref. [20], compound **7** was identified as oleuropein.

Compound **8**: Pale yellow oil; HRESIMS: *m*/*z* 303.2297 [M + Na]^+^ (calcd for C_18_H_32_O_2_Na, 303.2300). ^1^H NMR (CDCl_3_, 400 MHz): *δ* ppm 5.46–5.24 (m, 4H), 2.77 (t, *J* = 6.3 Hz, 2H), 2.34 (t, *J* = 7.5 Hz, 2H), 2.05 (q, *J* = 6.9 Hz, 4H), 1.62 (p, *J* = 7.2 Hz, 2H), 1.46–1.19 (m, 14H), 0.89 (t, *J* = 6.9 Hz, 3H). ^13^C NMR (CDCl_3_, 101 MHz) *δ* ppm 180.50, 129.94, 129.75, 127.96, 127.79, 33.99, 31.44, 29.50, 29.27, 29.07, 28.99, 28.95, 27.10, 27.07, 25.52, 24.54, 22.49, 13.91. Comparing these data with ref. [21], compound **8** was similar to the data of linolenic acid.

Compound **9**: Pale yellow oil; HRESIMS: *m*/*z* 317.2451 [M + Na]^+^ (calcd for C_19_H_34_O_2_Na, 317.2457). ^1^H NMR (CDCl_3_, 400 MHz): *δ* ppm 5.49–5.17 (m, 4H), 3.66 (s, 3H), 2.77 (t, *J* = 6.3 Hz, 2H), 2.30 (t, *J* = 7.5 Hz, 2H), 2.05 (dd, *J* = 6.8 Hz, 4H), 1.62 (t, *J* = 7.2 Hz, 2H),, 1.39–1.17 (m, 14H), 0.89 (t, *J* = 6.8 Hz, 3H). ^13^C NMR (CDCl_3_, 101 MHz): *δ* ppm 173.30, 129.60, 129.45, 127.68, 127.55, 50.74, 33.56, 31.20, 29.26, 29.03, 28.86, 28.80, 28.77, 26.84, 26.82, 25.26, 24.57, 22.25, 13.63. Comparing these data with ref. [22], compound **9** was similar to the data of methyl linoleate.

Compound **10**: White solid; m.p. 140 °C; HRESIMS: *m*/*z* 437.3755 [M + Na]^+^ (calcd for C_29_H_50_ONa, 437.3759). ^1^H NMR (CDCl_3_, 400 MHz): *δ* ppm 5.35 (m, 1H), 3.51 (m, 1H), 1.02 (s, 3H), 0.93 (d, *J* = 6.6 Hz, 3H), 0.85 (t, *J* = 7.1 Hz, 3H), 0.83 (t, *J* = 6.3 Hz, 3H), 0.81 (t, *J* = 7.1 Hz, 3H), 0.68 (s, 3H). ^13^C NMR (DMSO–*d_6_*; 101 MHz): *δ* ppm 140.72, 121.68, 71.75, 56.73, 56.01, 50.09, 45.79, 42.28, 42.24, 39.74, 37.22, 36.46, 36.11, 33.90, 31.87, 31.87, 31.60, 29.11, 28.22, 26.02, 24.27, 23.04, 21.05, 19.79, 19.37, 19.00, 18.75, 11.95, 11.83. Comparing these data with ref. [18], compound **10** was identified as *β*–sitosterol.

### 2.3. Biological Activities

#### 2.3.1. Determination of Antifungal Activities

All compounds were tested by the mycelium growth rate method [23] against six pathogenic fungi (*Rhizoctonia solani, Fusarium graminearum, Bipolaris maydis, Botrytis cinema, Sclerotium rolfsii,* and *Alternaria solani*) at the concentration of 0.2 mmol/L. The compounds were dissolved in acetone and the solutions were diluted with aqueous 0.1% Tween-80 and then added to sterile potato dextrose agar (PDA). After the PDA was solidified, the pathogenic fungi cakes (6 mm) were placed on the center of the culture plates and then incubated at 28 °C. The same concentration of the Carbendazim was used as the positive control, a common broad-spectrum fungicide. The same concentration of acetone and aqueous 0.1% tween-80 were used as negative control, repeating three times for each treat. When the mycelia of CK grew to 3/4 area of the diameter, it was measured by the cross-intersection method [24], and the inhibitory ratio was calculated by the Equation 1.

In order to accurately inspect the inhibitory effect of compound **2**, EC_50_ values were tested against six pathogens fungi. The compound was formed the stock solution of 0.8 mmol/L and divided into 6 gradients. Different gradient concentrations were 0.8 mmol/L, 0.6 mmol/L, 0.4 mmol/L, 0.2 mmol/L, 0.1 mmol/L, 0.05 mmol/L, respectively. The method of inoculating pathogenic fungi was equal to the method for testing the antifungal activities, and the method of measuring the circle mycelium was equal to Equation (1).
Relative inhibitory ratio (%) = [(CK − PT)/(CK − 6 mm)] × 100%(1)

CK is the diameter of the mycelium circle in the negative control group and PT is the diameter of the mycelium circle in the treatment group.

#### 2.3.2. Determination of Herbicidal Activities

All the compounds were tested the herbicidal activities against *Echinochloa crus-galli* and *Brassica napus* by the seed germination method [25]. The method of dispensing was equal to that in the experiment of antifungal activities. Subsequently, 5 mL of the solution was placed in 75 mm medium with double-layer filter paper when the seeds were emerge-germinating, and each culture dish was sow 12 seeds. The same concentration of Cyanazine was used as positive control, a common broad-spectrum herbicide. The seeds in the medium were germinated in the light incubator with the temperature 28 ± 1 °C, the humidity 80% rH, the luminance 1100 lux, and the photoperiod 14 h/day. When the root or stalk of the CK group grew to 40–50 mm, the measurement started. And the inhibitory ratios were calculated by Equation (2).

In order to test the EC_50_ value of compound **8** against *Barnyard grass*, 200 mL of 8 mmol/L stock solution was prepared and divided into five gradients, which were 8 mmol/L, 4 mmol/L, 2 mmol/L, 1 mmol/L, and 0.5 mmol/L, respectively. The method of sowing, cultivating, and measuring was equal to the above herbicidal activities experiment.
Relative inhibitory ratio (%) = [(CK − PT)/CK] × 100%(2)

CK is the root or stalk length of the negative control group and PT is the root or stalk length of the treatment group.

#### 2.3.3. Determination of Antibacterial Activities

In this experiment, all the isolates were examined by the filter paper dispersion methods [26] against *Erwinia carotovora, Pseudomonas syringae* and *Ralstonia solanacearum*. Each compound was prepared with 0.2 mmol/L of the solution. 90 mL beef extract peptone agar medium was made (peptone (m): NaCl (m): glucose (m): agar (m): deionized water (v) = 0.6:2:1:2:3.6:200), which were sterilized at 121 °C for 30 min, and poured into the 90 mm plate, 30 mL for each dish. When the medium was concretionary, a ring of activated bacteria was picked up by the inoculating loop, and parallel lines were drawn on the medium, and the sealing film was sealed in a 37 °C biochemical incubator for 1–2 days. 150 mL beef extract peptone liquid medium was made (peptone (m): NaCl (m): glucose (m): deionized water (v) = 0.6:2:1:2:200), which were sterilized at 121 °C for 30 min. When the bacterial suspension was formed, a ring of growing colonies was placed in liquid medium and placed at 37 °C with 150 r/min incubator for 24 h. Using saline to dilute the bacterial suspension to a concentration of 6 × 10^9^–6 × 10^10^ cfu/mL by double gradient dilution method. 0.2 mL of the diluted bacterial suspension were pipetted into 90 mL beef extract peptone agar medium (45–50 °C) and packed separately to solid. The filter paper was dipped into the extract solution for 3 s and pasted on the center of the delineated area of the bacteria-containing medium. The same concentration of tetracycline was used as positive control. Each treatment was repeated three times and placed in a 37 °C incubator for 24 h. The diameter of the inhibitory zone was measured by the cross method.

In order to ensure the validity of the results, the difference from two parallel measurements should not be greater than one step on the dilution scale. The experiments were conducted in duplicates. Each value is represented in terms of mean (n = 3) ± SD (Standard deviation). All activities were evaluated by statistical analysis. All statistical analysis was performed using EXCEL 2010 software. The log dose-response curves allowed determination of the EC_50_ for the fungi bioassay according to probit analysis. The 95% confidence limits for the range of EC_50_ values were determined by the least-square regression analysis of the relative growth rate (% control) against the logarithm of the compound concentration.

## 3. Results

### 3.1. Antifungal Activity

The result of antifungal activities was shown in Table 1. It can be seen the most of compounds have different levels of inhibitory effects against the tested fungi at the concentration of 0.2 mmol/L. Compounds **2**, **3**, and **4** have moderate inhibitory effects against all tested fungi, and compound **2** was the best, with the inhibitory effect of against *Sclerotium rolfsii* reaching 30.34 ± 8.43%, which was higher than positive control, Carbendazim, a broad-spectrum fungicide, whose inhibitory rate was 7.3 ± 4.59%. Comparing the bioassay results of compounds **1**–**5** belonging to the same skeleton of coumarin [27], the occurence of hydroxyl at C-8 could enhance the activity against *Sclerotium rolfsii* and the structure-activity relationship of these coumarin derivatives need be further studied later.

In addition, the inhibitory effect of compound **2** against *Alternaria solani* was also higher than the positive control, reaching 32.23 ± 0.49%. In order to further inspect the inhibitory activity of compound **2**, EC_50_ values were obtained, as shown in Table 2. These results showed that it could be used as a lead compound for broad-spectrum fungicide and had great potential development value and prospects.

### 3.2. Herbicidal Activity

The results of herbicidal activities were shown in Table 3. Among them, compounds **8** and **10** showed excellent inhibitory effects against roots of *Echinochloa crus-galli, which* were higher than 90%. And the inhibitory activitis against stalk of *E. crus-galli* were also excellent, reaching 78.07 ± 0.32% and 63.16 ± 0.32%, respectively, which were both higher than the positive control (40.39 ± 0.21%). According to the results of EC_50_ values, compound **8** showed outstanding inhibitory effects against the tested weeds (Table 4). It is interesting to note that compound **8** is not generally considered a bioactive ingredient, due to its widespread nature and little structural novelty. Nonetheless, previous studies on herbicidal and phytotoxic activities of linoleic acid and similar fatty acid derivatives, have been reported [28,29,30,31,32,33]. So, the structural modification of linoleic acid and its mechanism of herbicidal activity, remain to be valuable to further study.

### 3.3. Antibacterial Activity

Regrettably, no compound showed any antibacterial activities against the tested bacteria (*Erwinia carotovora, Pseudomonas syringae* and *Ralstonia solanacearum*). We speculated that we have not found the right bacteria, considering that all the selected bacteria are Gram-negative [34]. However, due to the limitations of research conditions, we failed to carry out further research on the broader selection and specific mechanism.

## 4. Conclusions

The genus *Fraxinus* contains 30 native species in China and only four species of them previously can be used as the origin of Chinese medicine ‘Qin-Pi’ which based on their morphological characteristics and folk medicinal efficacy. In addition, previous phytochemistry and chemotaxonomic research on *Fraxinus* have shown coumarins are their characteristic ingredients. In this work, chemical constituents from *F. hupehensis* were isolated systematically for the first time, and five of ten compounds obtained were coumarins. To our best knowledge, known coumarins **1**–**5** were isolated for the first time from the leaves of *F. hupehensis* and compound **5** was the first report in genus *Fraxinus*, which indicated that *F. hupehensis* and ‘Qin-Pi’ ′s species have a close relationship in term of chemical taxonomy.

Furthermore, All the isolated compounds were evaluated for antifungal, herbicidal and antibacterial activities. The results showed compound **2** deserves favourable effect for fungicide. The structural modification of **2**, belonging to the coumarin skeleton, and the structure-activity relationship of these coumarin derivatives have a very valuable prospect for the development of novel plant-derived pesticides in the future. More research is needed to probe into these activities and to explore the mechanisms of action of these active compounds.

## Figures and Tables

**Figure 1 biomolecules-10-00074-f001:**
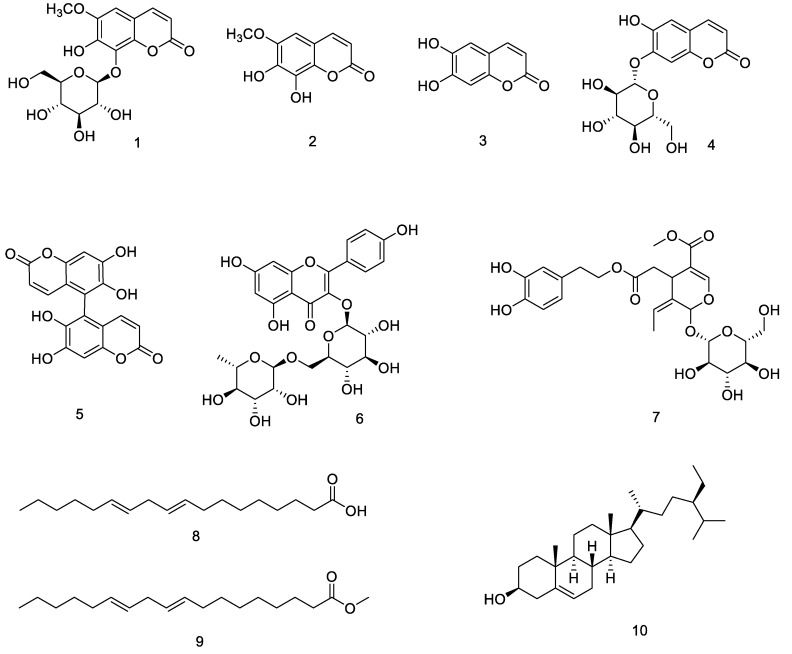
Chemical structures of **1**–**10** isolated from *F. hupehensis.*

**Table 1 biomolecules-10-00074-t001:** Inhibitory ratio of 10 compounds against six phytopathogenic fungi (Inhibitory ratio ± SD, %).

Compd.	*Rhizoctonia solani*	*Fusarium graminearum*	*Bipolaris maydis*	*Botrytis cinema*	*Sclerotium rolfsii*	*Alternaria solani*
**1**	−1.14 ± 0.99	1.28 ± 2.22	−0.72 ± 1.26	4.54 ± 1.07	9.46 ± 3.10	4.96 ± 0.07
**2**	36.37 ± 1.68	23.33 ± 2.89	37.39 ± 2.85	20.1 ± 1.82	30.34 ± 8.43	32.22 ± 2.09
**3**	27.78 ± 4.72	−1.33 ± 3.00	10.08 ± 2.57	6.44 ± 4.80	13.52 ± 6.26	0.79 ± 3.75
**4**	17.64 ± 1.44	1.26 ± 2.98	6.46 ± 2.08	11.67 ± 1.73	14.19 ± 0.16	0.81 ± 1.41
**5**	−1.63 ± 1.64	−2.37 ± 1.04	−2.48 ± 1.06	5.34 ± 3.29	−1.15 ± 3.42	2.1 ± 2.08
**6**	−3.24 ± 0.03	−0.06 ± 1.03	−0.31 ± 2.13	−0.11 ± 4.60	0.94 ± 4.81	1.4 ± 1.22
**7**	−1.62 ± 0.02	1.78 ± 1.79	−3.73 ± 1.85	0.69 ± 3.46	5.20 ± 3.14	3.52 ± 1.20
**8**	18.91 ± 2.32	−0.12 ± 2.71	0.31 ± 1.05	7.67 ± 3.21	2.58 ± 2.36	21.3 ± 2.13
**9**	18.94 ± 3.57	28.99 ± 1.93	0.37 ± 3.21	−0.85 ± 3.57	−1.67 ± 2.89	3.52 ± 1.21
**10**	32.43 ± 1.37	4.73 ± 1.00	−2.49 ± 2.16	−3.17 ± 2.75	−0.6 ± 2.49	3.49 ± 4.33
**Carbendazim**	100.00 ± 0.00	100.00 ± 0.00	24.23 ± 0.26	36.35 ± 3.38	7.3 ± 4.59	11.29 ± 2.56

Note: The values represent the mean ± SD of three individual observations.

**Table 2 biomolecules-10-00074-t002:** The EC_50_ values (mmol/L) of compound **2** against six pathogenic fungi.

Compd.	*Rhizoctonia solani*	*Fusa* *rium graminearum*	*Bipolaris maydis*	*Botrytis* *cinema*	*Sclerotium* *rolfsii*	*Alternaria* *solani*
**2**	0.33 ± 0.01	0.48 ± 0.02	0.31 ± 0.01	1.11 ± 0.02	0.50 ± 0.02	0.40 ± 0.02
**Carbendazim**	0.12 ± 0.00	0.13 ± 0.01	0.74 ± 0.01	0.32 ± 0.01	1.78 ± 0.01	1.41 ± 0.00

Note: The values represent the mean ± SD of three individual observations.

**Table 3 biomolecules-10-00074-t003:** Inhibitory ratio of 10 compounds against *Echinochloa crus-galli* and *Brassica napus* (Inhibitory ratio ± SD, %).

Compd.	*Echinochloa crus-galli*		*Brassica napus*	
	Root	Stalk	Root	Stalk
**1**	18.32 ± 0.29	4.13 ± 0.29	32.28 ± 0.82	−28.93 ± 1.76
**2**	40.36 ± 0.24	5.63 ± 0.36	13.26 ± 1.94	−45.62 ± 1.44
**3**	43.29 ± 0.23	17.56 ± 0.42	−25.53 ± 4.08	−33.06 ± 1.89
**4**	29.97 ± 0.27	18.66 ± 0.33	−16.85 ± 2.32	−51.37 ± 1.67
**5**	47.86 ± 0.27	13.14 ± 0.26	48.11 ± 1.22	−10.91 ± 0.37
**6**	9.95 ± 0.21	2.69 ± 0.18	−43.08 ± 4.16	−27.20 ± 0.38
**7**	4.61 ± 0.29	−7.92 ± 0.20	−29.04 ± 1.96	4.85 ± 0.27
**8**	96.71 ± 0.06	78.07 ± 0.32	29.82 ± 1.78	14.35 ± 0.39
**9**	64.85 ± 0.41	71.56 ± 0.34	−59.43 ± 4.03	−35.07 ± 0.78
**10**	91.43 ± 0.10	63.16 ± 0.32	28.54 ± 1.52	−10.77 ± 0.40
**Cyanazine**	66.52 ± 0.08	40.39 ± 0.21	48.83 ± 0.68%	6.78 ± 0.32

Note: The values represent the mean ± SD of three individual observations.

**Table 4 biomolecules-10-00074-t004:** The EC_50_ values (mmol/L) of compounds 8–10 against the root of *E. crus-galli.*

Compd.	Root	Stalk
**8**	1.16 ± 0.23	1.32 ± 0.27
**9**	1.28 ± 0.58	1.31 ± 0.46
**10**	1.33 ± 0.35	2.35 ± 0.98
**Cyanazine**	1.56 ± 0.44	2.84 ± 0.73

Note: The values represent the mean ± SD of three individual observations.

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
