# Peer review of "Chemical Constituents from Fraxinus hupehensis and Their Antifungal and Herbicidal Activities"

_biomolecules, 2020, doi:10.3390/biom10010074_

Round 1

Reviewer 1 Report

The current manuscript (biomolecules-671027) deals with the isolation of a series of constituents from extracts obtained from the leaves and bark of Fraxinus hupehensis, including data on their inhibitory effects towards phytopathogenic fungi and weeds. The study encompasses a certain degree of scientific novelty and relevance as it corresponds to the first report on the chemical constituents of the species, some of which exhibiting antifungal effects that appear to be relevant. However, and as detailed below, there are several issues that preclude the acceptance of the manuscript.

Firstly, English editing is mandatory as the manuscript is not only poorly but also uncarefully written, being one of the main factors hampering its final acceptance. Several examples can be found throughout the whole manuscript, as immediately evidenced in the Abstract:

Lines 19-20:Chemical compounds of Fraxinus hupehensis were systematically isolated and spectroscopic identification for the first time, and ten compounds were obtained, which were identified as…”.

The sentence is completely incoherent.

Lines 23-25:All the isolates except compounds 3 and 4 were isolated from the F. hupehensis for the first time, and the occurrence of compound 5 was the first report in the genus Fraxinus.”.

Once again, the sentence is unclear and incoherent.

There are several misspellings, including wrong taxonomic designations as in:

Lines 26-27: Revise “Selerotium” to “Sclerotium”; “Altemaria” / “Alternaria”.

Line 31: Revise “Echinochloa crusgalli” to “Echinochloa crus-galli”.

Line 34: Revise “Ervinia” to “Erwinia”.

Line 174: Revise “Fusaium” to “Fusarium”.

While the experimental procedures appear to have been properly conducted, the authors are requested to indicate the gradient used to fractionate fractions G (line 85) and O (line 87), as indicated for the remaining fractions.

Spectral data (1H and 13C-NMR) are in agreement with the structures of the isolated constituents; however, structures 6 and 10 (Figure 1) should be revised.  

It is also my personal opinion that the isolation of the fatty acids 8 and 9 as well as β-sitosterol (10) has a limited interest as they are ubiquitous in the plant Kingdom.

Section 2.3.1: Authors are requested to clarify if acetone interfered with the outcomes on the antifungal properties! Maximum percentage of acetone?

Section 3.1. Antifungal activity: Results on antifungal (antimicrobial) effects are commonly expressed in MIC.

Section 3.2. Herbicidal activity: In contrast with the antifungal effects of fraxetin (2), it is unlikely that the reported herbicidal properties of 8 and 10 are scientifically relevant due to the widespread distribution of such constituents.

Conclusions: The authors wrongly classify 8 and 9 as “phenolic acids” as they correspond to fatty acids.

Furthermore, the “conclusions” section corresponds to a mere repetition of the results, not conveying any concluding remarks.

Author Response

Reviewer 1:

Lines 19-20: “Chemical compounds of Fraxinus hupehensis were systematically isolated and spectroscopic identification for the first time, and ten compounds were obtained, which were identified as…”.The sentence is completely incoherent.

Response: Thank you for your good suggestions. We reedited the article carefully according to your advice. The corresponding revision has been done and were highlighted using yellow color.

Lines 23-25: “All the isolates except compounds 3 and 4 were isolated from the F. hupehensis for the first time, and the occurrence of compound 5 was the first report in the genus Fraxinus.”.Once again, the sentence is unclear and incoherent.

Response: Thank you for your good suggestions. The corresponding revision has been done.

There are several misspellings, including wrong taxonomic designations as in:

Lines 26-27: Revise “Selerotium” to “Sclerotium”; “Altemaria” / “Alternaria”.

Line 31: Revise “Echinochloa crusgalli” to “Echinochloa crus-galli”.

Line 34: Revise “Ervinia” to “Erwinia”.

Line 174: Revise “Fusaium” to “Fusarium”.

Response: Thank you for your good suggestions. We revised carefully according to your advice and all the changes were highlighted using yellow color.

While the experimental procedures appear to have been properly conducted, the authors are requested to indicate the gradient used to fractionate fractions G (line 85) and O (line 87), as indicated for the remaining fractions.

Response: Thank you for your good suggestions. We are very sorry for our carelessness and the corresponding revision has been done.

Spectral data (1H and 13C-NMR) are in agreement with the structures of the isolated constituents; however, structures 6 and 10 (Figure 1) should be revised.  

Response: Thank you for your good suggestions. The corresponding revision has been done.

It is also my personal opinion that the isolation of the fatty acids 8 and 9 as well as β-sitosterol (10) has a limited interest as they are ubiquitous in the plant Kingdom.

Response: Thank you for your good suggestions. First, since it was discovered as a new species in 1979 and called Living fossil or King of bonsai, there was almost no study on its phytochemisty. Therefore the research of its chemical constituents is valuable, even if they are just common known compounds. Second, our interest focus on discovering bioactive contituents from natural products, rather than only new compounds. So we sincerely hope that the revised manuscript will be worthy of publication in Biomolecules.

Section 2.3.1: Authors are requested to clarify if acetone interfered with the outcomes on the antifungal properties! Maximum percentage of acetone?

Response: Thank you for your good suggestions. In the antifungal activity test, the maximum concentration of acetone was 0.2%. And the difference in the average inhibition rate between the acetone group and the blank control group was less than 10%, so the affect of acetone on the antifungal activity could be ignored.

Section 3.1. Antifungal activity: Results on antifungal (antimicrobial) effects are commonly expressed in MIC.

Response: Thank you for your good suggestions. We have searched for a lot of related literature on antimicrobials and pesticides, and find that MIC are commonly used in medical/clinical antifungl or antibacterial activity. In the antifungal activity of results of pesticides, inhibitiory ratio of primary screening also often used and evaluated, combined with half maximal effective concentration (EC50). And we found two relevant articles (1.Design, synthesis and antifungal activity of amide and imine derivatives containing a kakuol moiety, Bioorganic & Medicinal Chemistry Letters, 2020, 30, 126774; 2. (R)-2-Phenyl-4,5-Dihydrothiazole-4-Carboxamide Derivatives Containing a Diacylhydrazine Group: Synthesis, Biological Evaluation, and SARs, Molecules, 2019,24,4440.). So we sincerely hope that the revised manuscript will be worthy of publication in Biomolecules.

Section 3.2. Herbicidal activity: In contrast with the antifungal effects of fraxetin (2), it is unlikely that the reported herbicidal properties of 8 and 10 are scientifically relevant due to the widespread distribution of such constituents.

Response: Thank you for your good suggestions. Although these two compounds are abundant in plants, we want to study phytochemistry on Fraxinus hupehensis systematically and evaluate these isolated compounds as more activities as possible, in order to allow us perform a more comprehensive activity analysis of extract and its chemical components.

So we sincerely hope that the revised manuscript will be worthy of publication in Biomolecules.

Conclusions: The authors wrongly classify 8 and 9 as “phenolic acids” as they correspond to fatty acids.

Response: Thank you for your meaningful suggestions. The corresponding revision has been done.

11.Furthermore, the “conclusions” section corresponds to a mere repetition of the results, not conveying any concluding remarks.

 Response: Thank you for your well-meaning reminder. We resummarize and reedited the conclusion. Subsequent work will work on the mechanism of action of these substances.

Reviewer 2 Report

lines 19-20 must be revised

keywords: correct "chemcial"

text in general: bibliographic references are not correctly reported

line 46: change to "called Living"

line 58: change to "have shown"

fungal and bacterial strains: are they purchased from a commercial supplier (ATCC) or are they experimentally isolated?

line 217: delete "it was"

lines 221-222 must be revised

line 226: I suggest "were pipetted"

line 227: delete "it was"

line 235: change to "was shown" and "seen that most of"

lines 239 and 242 and 254: change to "reaching"

line 240: change to "fungicide, whose inhibitory"

line 242: change to "was also higher"

line 245: change to "as shown in Table 2"

line 246: the sentence on coumarin is unclear

line 255: change to "higher than"

Table 4 caption: change to "compounds"

paragraph 3.3: bacteria not bacterias

line 267: delete "of important substances"

line 292: correct "neede"

line 299: change to "Hsiang who revised"

line 300: correct "intruduction"

Author Response

Reviewer 2:

lines 19-20 must be revised

 Response: Thank you for your good suggestions. The corresponding revision has been done.

keywords: correct "chemcial"

 Response: Thank you for your good suggestions. The corresponding revision has been done.

text in general: bibliographic references are not correctly reported

 Response: Thank you for your good suggestions. The corresponding revision has been done.

line 46: change to "called Living"

 Response: Thank you for your good suggestions. The corresponding revision has been done.

line 58: change to "have shown"

 Response: Thank you for your good suggestions. The corresponding revision has been done.

fungal and bacterial strains: are they purchased from a commercial supplier (ATCC) or are they experimentally isolated?

 Response: Thank you for your good suggestions. All six tested phytopathogenic fungi (Rhizoctonia solani, Fusarium graminearum, Bipolaris maydis, Botrytis cinema, Selerotium rolfsii, and Altemaria solani) and three tested bacterias (Ervinia carotovora, Pseudomonas syringae and Ralstonia solanacearum) were provided by the Institute of Pesticide Research, Yangtze University.

line 217: delete "it was"

 Response: Thank you for your good suggestions. The corresponding revision has been done.

lines 221-222 must be revised

 Response: Thank you for your good suggestions. The corresponding revision has been done.

9.line 226: I suggest "were pipetted"

 Response: Thank you for your good suggestions. The corresponding revision has been done.

10.line 227: delete "it was"

 Response: Thank you for your good suggestions. The corresponding revision has been done.

11.line 235: change to "was shown" and "seen that most of"

 Response: Thank you for your good suggestions. The corresponding revision has been done.

12.lines 239 and 242 and 254: change to "reaching"

 Response: Thank you for your good suggestions. The corresponding revision has been done.

13.line 240: change to "fungicide, whose inhibitory"

 Response: Thank you for your good suggestions. The corresponding revision has been done.

14.line 242: change to "was also higher"

 Response: Thank you for your good suggestions. The corresponding revision has been done.

15.line 245: change to "as shown in Table 2"

 Response: Thank you for your good suggestions. The corresponding revision has been done.

16.line 246: the sentence on coumarin is unclear

 Response: Thank you for your good suggestions. The corresponding revision has been done.

17.line 255: change to "higher than"

 Response: Thank you for your good suggestions. The corresponding revision has been done.

18.Table 4 caption: change to "compounds"

 Response: Thank you for your good suggestions. The corresponding revision has been done.

19.paragraph 3.3: bacteria not bacterias

 Response: Thank you for your good suggestions. The corresponding revision has been done.

20.line 267: delete "of important substances"

 Response: Thank you for your good suggestions. The corresponding revision has been done.

21.line 292: correct "neede"

 Response: Thank you for your good suggestions. The corresponding revision has been done.

22.line 299: change to "Hsiang who revised"

 Response: Thank you for your good suggestions. The corresponding revision has been done.

23.line 300: correct "intruduction"

 Response: Thank you for your good suggestions. The corresponding revision has been done.

Reviewer 3 Report

Zhao et al. have achieved the identification of ten compounds from Fraxinus hupehensis, some of them with antifungal and herbicidal properties. 

The finding of new antifungal/herbicidal compounds from natural sources is interesting, they got them form leaves and bark, with relatively good EC50 values.

However, the methodology should be further explained and the inhibition percentage could be calculated additionally by biomass reduction, which gives additional and more accurate information.  English should be improved and species name is usually written with mistakes, please correct that throughout the paper.

Please find the paper attached with several comments in order to improve it.

Author Response

Reviewer 3:

1.Zhao et al. have achieved the identification of ten compounds from Fraxinus hupehensis, some of them with antifungal and herbicidal properties. The finding of new antifungal/herbicidal compounds from natural sources is interesting, they got them form leaves and bark, with relatively good EC50 values. However, the methodology should be further explained and the inhibition percentage could be calculated additionally by biomass reduction, which gives additional and more accurate information.  English should be improved and species name is usually written with mistakes, please correct that throughout the paper.

Response: Thank you for your good suggestions. The method for determining the EC50 value of fungicidal activities were described in the part of materials and methods and these methods were based on the reference. The values of EC50 were calculated after detecting the fungicidal activity of 5 gradient concentrations. And we have obtained the value±SD of three individual observations for the tested isolates and all fungicidal activities were evaluated by statistical analysis. The corresponding revision has been done.

2.Please find the paper attached with several comments in order to improve it.

Response: Thank you for your good suggestions. We have polished and reedited the manuscript. The corresponding revision has been done according to your advice.

Round 2

Reviewer 1 Report

While addressing some of the issues raised by the reviewers, the authors of the manuscript biomolecules-671027 were unable to sufficiently improve the manuscript in order to be accepted in Biomolecules.

As in the previous version of the manuscript, extensive editing of English language and style is required, authors being strongly advised to have the manuscript revised by an English editing service or a native speaker. Examples that the manuscript was insufficiently improved can be found in:

Lines 34-35:The Structures of these isolated constituents were characterized identified by 1H NMR, 13C NMR and HRMS.”.

Lines 38-39: “Antifungal activity evaluation tests showed that compound 2 exhibited significant inhibition inhibitory effects against …”.

Lines 46-47: “However, all none of the compounds proved to be active had no inhibitory activity against…”.

Lines 54-56: “…and 30 native species in China., Many many of which species of them have been commonly used as in Traditional Chinese Medicine.”

Several typos can still be found throughout the whole manuscript as in lines 270 (“inhibiton”), 273 (“Selerotium”) and 309 (“characterstic”).

Several keywords are unspecific and inadequate i.e. “methanol extract” and “chemical constituents”.

As in the original version of the manuscript, authors included a series of assumptions that appear to reflect a personal view rather than a scientifically coherent premise conveniently corroborated by references, as well as decontextualized content as in:

Lines 61-62: Authors claim that the species under study is strongly resistant to “pests and diseases”, without any further contextualization or reference supporting such claims.

Lines 220, 288-289, Table 3:Barnyard grass” refers to a common name rather than a taxonomic designation, being unclear if the authors solely refer to Echinochloa crus-galli.

Lines 280-283: It is mentioned that fraxetin (2) displayed “low toxicity and drug resistance”, being unclear if the authors refer to their own data or previous studies. Similarly, while being also referred that coumarins exhibit “anti-oxidation, anti-inflammatory, anti-viral, anti-tumor, anti-hyperglycemia activities”, there are no experimental data or references supporting those effects. Furthermore, is there any relationship between those effects and the recorded antifungal properties against plant pathogenic fungi?!

Lines 300-304: As none of the isolated compounds proved to efficiently inhibit bacterial growth, discussion on the interference on TonB transporters is out of context, as the authors did not deliver any experimental data on the potential interference.

Lines 316-317: Authors refer that “Plentiful coumarins indicated that F. hupehensis and “Qin-Pi”’s species have a close relationship.

What type of relationship? Taxonomic? Chemotaxonomic? Which of the species commonly classified as Q”in-Pi”?

As mentioned on my previous report, I still fuel my opinion that linoleic acid (compound 8) has a more than limited potential to be developed as an herbicide, not only due to its widespread distribution but also due to its low biological interest.

Finally, and in agreement with the Instructions for Authors, references must be solely numbered in order of appearance in the text and placed in square brackets.

Author Response

Reviewer 1:

Lines 34-35:The Structures of these isolated constituents were characterized identified by 1H NMR, 13C NMR and HRMS.”.

Response: Thank you for your good suggestions. The corresponding revision has been done and were highlighted using yellow color.

Lines 38-39: “Antifungal activity evaluation tests showed that compound 2 exhibited significant inhibition inhibitory effects against …”.

Response: Thank you for your good suggestions. The corresponding revision has been done and were highlighted using yellow color.

Lines 46-47: “However, all none of the compounds proved to be active had no inhibitory activity against…”.

Response: Thank you for your good suggestions. We revised carefully according to your advice and all the changes were highlighted using yellow color.

Lines 54-56: “…and 30 native species in China., Many many of which species of them have been commonly used as in Traditional Chinese Medicine.”

Response: Thank you for your good suggestions. The corresponding revision has been done and were highlighted using yellow color.

Several typos can still be found throughout the whole manuscript as in lines 270 (“inhibiton”), 273 (“Selerotium”) and 309 (“characterstic”).

Response: Thank you for your good suggestions. The corresponding revision has been done.

Several keywords are unspecific and inadequate i.e. “methanol extract” and “chemical constituents”.

Response: Thank you for your good suggestions. We revised these keywords and were highlighted using yellow color.

As in the original version of the manuscript, authors included a series of assumptions that appear to reflect a personal view rather than a scientifically coherent premise conveniently corroborated by references, as well as decontextualized content as in: Lines 61-62: Authors claim that the species under study is strongly resistant to “pests and diseases”, without any further contextualization or reference supporting such claims.

Response: Thank you for your good suggestions. We have polished the text again and the relevant reference has been added. Please see Ref.3

Lines 220, 288-289, Table 3: “Barnyard grass” refers to a common name rather than a taxonomic designation, being unclear if the authors solely refer to Echinochloa crus-galli.

Response: Thank you for your good suggestions. We are very sorry for our carelessness and the name was unified with Echinochloa crus-galli in the doc.

Lines 280-283: It is mentioned that fraxetin (2) displayed “low toxicity and drug resistance”, being unclear if the authors refer to their own data or previous studies. Similarly, while being also referred that coumarins exhibit “anti-oxidation, anti-inflammatory, anti-viral, anti-tumor, anti-hyperglycemia activities”, there are no experimental data or references supporting those effects. Furthermore, is there any relationship between those effects and the recorded antifungal properties against plant pathogenic fungi?!

Response: Thank you for your good suggestions. We polished the text again and focused on the discussion of fraxetin’s antifungal activity. And the relevant reference has been added. Please see Ref. 27.

Lines 300-304: As none of the isolated compounds proved to efficiently inhibit bacterial growth, discussion on the interference on TonB transporters is out of context, as the authors did not deliver any experimental data on the potential interference.

Response: Thank you for your meaningful suggestions. We reorganized the discussion and the corresponding revision has been done.

Lines 316-317: Authors refer that “Plentiful coumarins indicated that F. hupehensis and “Qin-Pi”’s species have a close relationship.”What type of relationship? Taxonomic? Chemotaxonomic? Which of the species commonly classified as” Qin-Pi”?

 Response: Thank you for your well-meaning reminder. There are four species commonly classified as “Qin-Pi” in Traditional Chinese Medicine, including F. rhynchophylla Hance, F. chinensis Roxb, F. szaboana Lingelsh and F. stylosa Lingelsh. These plants are related to each other in chemical taxonomy. Previous phytochemistry on Fraxinus have shown coumarins are their main ingredients and the corresponding references are added.

As mentioned on my previous report, I still fuel my opinion that linoleic acid (compound 8) has a more than limited potential to be developed as an herbicide, not only due to its widespread distribution but also due to its low biological interest.

Response: Thank you for your good suggestions. We respect and value your opinion very much. To this end, we have reviewed the experimental process, checked the experimental procedures and results, to ensure the authenticity and accuracy of the existing experimental results. In addition, we also have consulted lots of literature and found that linoleic acid and related fatty acids had herbicidal activity in several scientific papers. Therefore, we do not believe that the herbicidal activity of linoleic acid is uncommon. Finally, we think the structural modification of linoleic acid and its mechanism of herbicidal activity, remain to be valuable to further study in the future. The corresponding references are added and please see Ref.28-33.

Finally, and in agreement with the Instructions for Authors, references must be solely numbered in order of appearance in the text and placed in square brackets.

Response: Thank you for your good suggestions. The references have been numbered again in order of appearance in the text and placed in square brackets. The corresponding revision has been done.

Reviewer 3 Report

The paper has been improved, anyway there is an item which has not been fulfilled. The conclusion should not be a single abstract, the authors should focus on the biological significance of their findings. Currently, they make a brief explanation what they have made in this study. So, this section must be improved.

Additionally, in the line 269 seems to be a grammar mistake: "It can seen the most of"

Best regards

Author Response

Reviewer 3:

The paper has been improved, anyway there is an item which has not been fulfilled. The conclusion should not be a single abstract, the authors should focus on the biological significance of their findings. Currently, they make a brief explanation what they have made in this study. So, this section must be improved.

Response: Thank you for your good suggestions. We resummarize and reedited the conclusion according to your advice. The corresponding revision has been done.

Additionally, in the line 269 seems to be a grammar mistake: "It can seen the most of".

Response: Thank you for your good suggestions. The corresponding revision has been done according to your advice.